# Human-like Decision Making for Autonomous Vehicles at the Intersection Using Inverse Reinforcement Learning

**DOI:** 10.3390/s22124500

**Published:** 2022-06-14

**Authors:** Zheng Wu, Fangbing Qu, Lin Yang, Jianwei Gong

**Affiliations:** 1Vehicle Engineering Department, Inner Mongolia Technical College of Mechanics and Electrics, Huhehot 010070, China; 1071006@nmgjdxy.edu.cn; 2School of Mechanical and Engineering, Beijing Institute of Technology, Beijing 100081, China; 3120200369@bit.edu.cn (F.Q.); gongjianwei@bit.edu.cn (J.G.)

**Keywords:** human-like interactive, intelligent vehicles, inverse reinforcement learning, semi-Markov model

## Abstract

With the rapid development of autonomous driving technology, both self-driven and human-driven vehicles will share roads in the future and complex information exchange among vehicles will be required. Therefore, autonomous vehicles need to behave as similar to human drivers as possible, to ensure that their behavior can be effectively understood by the drivers of other vehicles and be more in line with the cognition of humans on driving behavior. Therefore, this paper studies the evaluation function of human drivers, using the method of inverse reinforcement learning, aiming for the learned behavior to better imitate the behavior of human drivers. At the same time, this paper proposes a semi-Markov model, to extract the intentions of surrounding related vehicles and divides them into defensive and cooperative, leading the vehicle to adopt a reasonable response to different types of driving scenarios.

## 1. Introduction

With the development of autonomous driving technologies, autonomous vehicles are considered an important part of the future transportation system, in terms of environmental perception, decision making, path planning and motion control. Google’s unmanned vehicles have traveled millions of kilometers, while in the automatic driving mode. Waymo company, its subsidiary, has planned to enter the driverless taxi industry. In addition, Tesla and Uber also have great potential in the field of autonomous driving. Tesla has developed its own automatic driving system, called Autopilot and applied it to a model S vehicle, to achieve the function of automatic driving [1]. In China, many IT giants have also joined the venture, such as Baidu, Alibaba and Huawei, injecting new vitality into the research and development of driverless cars, with their technology in the field of artificial intelligence. Baidu Apollo’s automatic driving test mileage has exceeded 25 million kilometers travel distance. Baidu made available its unmanned vehicle service to the public, in Shougang Park in May 2021, taking the lead in entering a new stage of unmanned driving. Following, an unmanned driving test was conducted on the open road, in Yizhuang. Baidu’s driverless transport service has also gradually expanded, from the closed park to people’s daily travel needs. However, on the public roads with complex traffic conditions, the overemphasis on the “correctness” of the perception control mapping process [2] often causes the driving behavior [3] with a certain social intelligence to be overlooked, which may lead to serious safety problems. For example, in 2016, Google’s driverless car collided with a municipal bus, because the driverless strategy misjudged the intention of the bus to give way [4]. This judgment obviously does not conform to the driving behavior of human driving and does not realize the social aspect of human drivers, when making decisions [5]. That is, the current driverless strategy does not consider the driving behavior and decision-making intelligence, based on the long-term accumulated driving experience of humans. In the future, intelligent vehicles with autonomous driving ability will travel the roads alongside with vehicles driven by human drivers [6], in a mixed traffic scenario that will exist for a long time [1,7,8,9]. In order to reduce the discomfort of human drivers, in the face of intelligent vehicles with automatic driving function, it is necessary for autonomous vehicles to learn the behavior and logic of human drivers. Human beings have excellent scenario generalization, skill learning and emergency handling abilities. Autonomous vehicles (AVs) are publicly acceptable only if their driving behavior is comprehensible and comparable to that of human drivers [10,11,12]. Therefore, the study of the driving behavior of human drivers is an important aspect of intelligent driving vehicle research, towards making the automatic driving vehicles understand human driving mode, drive like humans and last enhancing people’s acceptance of AVs.

Human driving behavior modeling mainly follows two approaches: model-based method and data-based method. Model-based methods require a priori knowledge, which can be obtained from experiments, physics, and so on, while they are usually divided into perception, decision-making and execution modules [7]. Autonomous vehicle driving basic tasks (such as steering, speed control, overtaking and obstacle avoidance) are based on model predictive control, Markov chain modeling and adaptive control. These model-based methods are usually not intended to directly imitate human driving behavior, while they are used in cases, such as presented in [1,13,14,15].

The Markov Decision Process (MDP) is one of the common methods to describe the human-like decision-making process [16]. Based on MPD, a kind of human longitudinal decision model is designed to solve the speed planning problem at a signalized intersection [17]. However, this model does not consider horizontal decision-making. In [1], the Hidden Markov Model (HMM) is used to describe the lane changing intention. In [18], the Partial Observable Markov Decision Process (POMDP) is used to make decisions at unsignalized intersections, realizing human-like driving.

The data-driven human-like learning algorithm does not need any system information, such as a mathematical model and system parameters [19]. This method has been widely used in human-like driving and decision making [20,21,22], especially the deep neural network approach, usually combined with reinforcement and imitation learning, has been used to model various driving tasks in different environments [23]. A human-like longitudinal driving model for AVs is established, by using reinforcement learning (RL) [24]. Deep reinforcement learning is also used to design human-like car-following models [25]. In [26], implementing double Q-learning, human-like speed control can be realized. In [27], a method of extracting driving behavior trades from human expert drivers is proposed and applied to automatic agents, in order to reproduce active driving behavior. In [28], a more accurate, comfortable and humanized driving model is obtained, by training and evaluating a Drive360 data set. In [21], a learning method of motion planning, based on cost parameters, derived from natural driving data, is proposed. Lu et al. [29] propose a reinforcement learning system to learn driver behavior and realize human-like control. In addition, the end-to-end method is also one of the commonly used data-driven methods. An interpretable end-to-end deep reinforcement learning approach for autonomous driving is proposed in [30], capable of handling complex urban scenarios. A method capable of learning an end-to-end autonomous driving control policy using sparse rewards is proposed in [31]. Ref. [32] improves the performance and generalization of end-to-end autonomous driving by leveraging scene understanding with deep learning and multimodal sensor fusion techniques. Ref. [33] proposes an approach to improve the stability of end-to-end autonomous driving based on deep reinforcement learning. Ref. [34] summarizes the end-to-end approach to autonomous driving so that readers can understand the end-to-end approach.

Inverse Reinforcement Learning (IRL) is an important method of human-like learning. IRL refers to an algorithm that reversely deduces the reward function, on the premise of providing a strategy or an expert example, so that the agent can learn how to make decisions on complex issues, through expert demonstration. Inverse Reinforcement Learning (IRL) can deal with the problems related to learning driving style from expert demonstration [35]. Then, the IRL method is developed into maximum entropy IRL [36], so the driving trajectories of expert examples on the highway can be learned [37]. These methods are to obtain the weight of the reward function, through feature matching between the best trajectory of the reward function and the expert demonstration data. Levine and Koltun [38] introduced an approach that facilitates the use of local optimal demonstrations for inverse reinforcement learning. Naumann et al. [39] aim to investigate to what extent cost functions are suitable for interpreting and imitating human driving behavior. They focus on how cost functions differ from each other, in different driving scenarios. Huang et al. [40] used a driving model, based on internal reward function, to simulate human decision-making mechanism, inferred the reward function from real human driving data.

In [41], a large number of literature surveys are conducted, referring to many literatures related to end-to-end and IRL methods, and the respective characteristics of the two methods are listed. The end-to-end method has the ability to self-optimize, does not require prior knowledge, and has a black-box nature. End-to-end methods mostly use behavioral cloning, which is simpler than other methods. However, the end-to-end method also has shortcomings such as insufficient datasets that can be used [42], poor generalization ability, inability to take into account task security issues, and lack of standard evaluation methods. The IRL method uses expert demonstration data to train the system, has good generalization ability, and the required human driver data is easy to collect. However, the IRL method also has certain shortcomings: the computational complexity of the reward function is high, and the reward function has a certain ambiguity, and the amount of data required by the IRL method is also relatively large.

At present, the related work on human-like driving has been developed to a certain extent, but the scenes involved in these works are mostly straight driving roads, and rarely involve intersection scenes. The intersection scene is the main component of traffic, and it is also the road section with frequent traffic accidents, so the research on the intersection scene is very important. This paper uses the inverse reinforcement learning algorithm to simulate the driving behavior of human drivers, in the intersection scenario, so as to improve the understandability of the behavior of autonomous vehicles, when passing through an intersection.

The rest of this paper is organized as follows: Section 2 introduces the decision-making model in an intersection scenario and the related content of inverse reinforcement learning. Section 3 includes the verification in a Carla simulation environment. Section 4 carries out the experiment in an actual traffic scenario. Section 5 summarizes the methods used in this paper and points out the shortcomings.

## 2. Intersection Decision Framework

### 2.1. Humanoid Decision-Making Process at an Intersection

Driverless decision-making systems need algorithms that can deal with complex situations. In particular, urban scenarios, such as intersections with multiple pedestrians, traffic lights, cars and bicycles, are a huge challenge, even for human drivers. In order to make the autonomous vehicle run safely and reliably in such a complex and realistic environment, various technical challenges need to be considered and verified repeatedly in real and virtual scenarios.

As shown in Figure 1, studies of the decision-making process of human drivers, in dynamic and complex environments, contribute to building a decision-making model for uncertainty. Human generated data are used to optimize the model and adjust the parameters, so that the output behavior of the model is humanlike. Furthermore, the formulation of reasonable evaluation indexes and training models can improve the performance of agents. Finally, the simulation experiment can provide a safe test environment. Based on simulation experiments, the model can be continuously optimized and the generalization ability of the decision model can be improved.

### 2.2. Scenario Evaluation

The main purpose of this study is to determine whether there are safety issues between the test vehicle and the opposite moving vehicle. After the evaluation of the driving track of the opposite moving vehicle, there is the question of coincidence area of the global planned trajectory of the test vehicle, along the direction of the driving track of the opposite moving vehicle. For example, as shown in Figure 2, the upper left vehicle is the test vehicle (ego vehicle), the lower right vehicle is the opposite moving vehicle, and the coincidence area of red track and the green track is the potential collision area. If there is an overlapping area, the opposite moving vehicle is the vehicle with potential collision risk. The remaining vehicles are classified as irrelevant vehicles and they are not evaluated, because it is considered that such vehicles will not pose a potential threat to the test vehicle. This is a highly computational method, especially when there are many interactive vehicles, at the intersection, thus, the effective screening of the relevant vehicles can improve the calculation efficiency. Later, the possibility of collision will be evaluated in a reliable way.

### 2.3. Time to Collision (TTC)

This paper uses TTC to quantitatively divide the current scenario. TTC is mainly used as the standard to evaluate the potential threat in a collision mitigation system, because it represents the relationship between spatial proximity and velocity difference. First, the time when other vehicles enter the area of possible conflict is used to make a preliminary division of the moving vehicles. A time instance is set as the threshold of the possible conflict area, while the surrounding vehicles are divided into dangerous situation, radical situation and safety situation. Different driving strategies are implemented, according to the respective classification into the various dangerous situations.

Noh [43] mentioned the division of dangerous scenarios, using collision time (TTC) as the standard, to evaluate the potential threat, at unsignalized T-intersections [44,45,46].

As shown in Table 1, there is a strong correlation between the TTC and the actual braking time compared with the speed of the vehicle and the distance of the vehicle from the conflict scene. In order to verify the proposed hypothesis in this paper, a linear fitting analysis is performed, as shown in Figure 3. Considering the relationship between the braking time of the real vehicle and the ratio between the distance between the vehicle and the virtual conflict point and the speed of the vehicle, the following relationship is obtained:$$T_{break}=2.737\times (d/v)+0.1512$$
where
d: Distance between the test vehicle and the virtual collision point;v: Speed of the test vehicle.


According to the international standard ISO22 179:2009, a threshold value of 3 s is assigned. Specifically, when the time of future collision, as calculated according to the current state of the vehicle, is greater than 3 s, the current situation is regarded as safe, whereas when it is less than or equal to 3 s, the current situation is regarded as dangerous.

### 2.4. Semi-Markov Model

The problem formulation in this study is similar to MAXQ method [47]. In this paper, the mathematical expression of time abstraction is constructed according to the Semi-Markov Decision Process (SMDP) [48]. SMDP is an abstract form of Markov Decision Process (MDP). Contrary to MDP, which can only perform operations in a single time step, the operations in the SMDP framework can last for a variable period of time. In order to better describe various forms of behavior with obvious differences, a semi-Markov model is used [49]. The SMDP framework proposed in this paper is shown in Algorithm 1.

**Algorithm 1** SMDP Framework
In the planning window **t**:
**If** (vtrtkt+1/2atrtkt2<dtrik(ck) **and** vtrtk−atrtkt≤0)
**Then** Execute cooperative strategy**If** (dtrik(ck)<vtrtkt+1/2atrtkt2<dtrik(ck)+W+L)
**Then** Execute defensive strategies**Else** Execute general strategy

In this paper, a hierarchical strategy gradient method is proposed, which uses the time abstract formula of Semi-Markov Decision Process (SMDP), to solve the strategy. This method is applied to the scenario of autonomous vehicle at an intersection. The vehicle can exhibit two different types of behavior (such as defensive behavior and cooperative behavior), while its sub actions (such as acceleration and deceleration) should follow the corresponding behavior. The real data verification and simulation results in this paper show that, the proposed method can lead to correct decisions in the scenario of interaction with other vehicles at an intersection, and the produced behavior is similar to real human response. On the contrary, using MDP solution method does not achieve good experimental results, while there is also a large variance. In addition, the layered strategy method can be used in similar modules in the future, to cover more scenarios [49].

In the process of drafting a strategic solution, this paper adopts the approach of policy search, within the set planning window. In the environment of an intersection, the length of planning window is different, due to the different environment. In dangerous scenarios, the planning window is set to 3 s, while in the non-hazardous scenario, the planning window is selected to be 8 s long. In the process of strategic solution, the selection of the prediction step will also affect the quality of the final result. If the vehicle is in a dangerous situation, the prediction step is set at 0.5 s. When the vehicle is in a non-dangerous state, the prediction step is selected to be 1 s. In the solving process of longitudinal decision-making, based on this model and according to the current motion state of the autonomous vehicle and the prediction state of the surrounding vehicles, the candidate strategy set *P* is generated, under the given prediction period *H* and prediction step size. Each candidate strategy is calculated according to the evaluation function, while the one with the largest reward value is output as the future driving action of the vehicle.

### 2.5. Maximum Entropy Inverse Reinforcement Learning

The reward function of traditional reinforcement learning is often set artificially, and there is a certain irrationality. Inverse reinforcement learning [50] considers that the strategy of the expert example is optimal or close to the optimal, and then learns the reward function from the expert example, imitating the driving strategy of the expert, which is a human-like learning method.

The inverse reinforcement learning algorithm used in this paper, is based on human suboptimal behavior. This paper adopts the method of probabilistic graph modeling to solve the ambiguity of path selection. At the same time, based on the principle of maximum entropy, the reward function is restored [36].

In practical problems, because the state of vehicle is a continuous space, the maximum entropy method will lead to an extremely large solution space and function non-convergence. To solve this problem, a probability-based inverse optimal control algorithm is introduced in this paper. The implementation of Inverse Reinforcement Learning (IRL) algorithm restores the unknown reward function in Markov Decision Process (MDP), based on the expert demonstration of corresponding strategy. This reward function can be used for apprenticeship, reinforcement learning, and so on, to extend the behavior of experts to new situations, or to infer the behavior goals of experts.

Considering a continuous state $s={\left(s_{1},\ldots ,s_{T}\right)}^{T}$, a continuous action $a={\left(a_{1},\ldots ,a_{T}\right)}^{T}$ and discrete time, this kind of task is characterized by a dynamic function F. This paper defines it as: $F(s_{t-1},a_{t})=s_{t}$. At the same time, the reward function is also defined as $r(s_{t},a_{t})$. Given the initial state $s_{0}$, the optimal action is determined by $a=\underset{a}{\arg\max}\sum_{t}r(s_{t},a_{t})$.

In this study, the maximum entropy model, closely related to the linear Markov model, is used for modeling. According to this model, the probability of selecting action a is directly proportional to the index of rewards, as encountered in this process:(1)$$P(a|s_{0})=\frac{1}{Z}\exp(\sum_{t}r(s_{t},a_{t}))$$
where Z is the partition function.

An approximation of Equation (1), as given below, allows efficient learning in high-dimensional continuous domains. In addition to breaking the exponential dependence on dimension, this approximation needs only one example of approximate local optimal, instead of an example trajectory being globally optimal.

It is assumed that experts perform local optimization, rather than global planning, when selecting action a. Compared to the global optimality requirement, local optimization shows much less limitation. Using r(a) to denote the sum of rewards along the path, Equation (1) can be written as:(2)$$P(a|s_{0})=e^{r(a)}{\left[\int e^{r(\tilde{a})}d\tilde{a}\right]}^{-1}$$

This probability is approximated by the second-order Taylor expansion of the reward, near the action:(3)$$r(\tilde{a})\approx r(a)+{\left(\tilde{a}-a\right)}^{T}\frac{\partial r}{\partial a}+\frac{1}{2}{\left(\tilde{a}-a\right)}^{T}\frac{\partial^{2}r}{\partial a^{2}}(\tilde{a}-a)$$

Denoting the gradient $\frac{\partial r}{\partial a}$ as g and the Hessian $\frac{\partial^{2}r}{\partial a^{2}}$ as H, the approximation to Equation (2) is given by:(4)$$\begin{array}{l} P\left(a|s_{0}\right)\approx e^{r(a)}{\left[\int e^{r(a)+{\left(\tilde{a}-a\right)}^{T}g+\frac{1}{2}{\left(\tilde{a}-a\right)}^{T}H(\tilde{a}-a)}d\tilde{a}\right]}^{-1}\\ =e^{\frac{1}{2}g^{T}H^{-1}g}{\left|-H\right|}^{\frac{1}{2}}{\left(2\pi \right)}^{-\frac{da}{2}} \end{array}$$

The approximate log likelihood is obtained as:(5)$$ℒ=\frac{1}{2}g^{T}H^{-1}g+\frac{1}{2}\log|-H|-\frac{d_{a}}{2}\log2\pi$$

The gradient of the reward parameter θ is:(6)$$\frac{\partial ℒ}{\partial \theta }=h^{T}\frac{\partial g}{\partial \theta }-\frac{1}{2}h^{T}\frac{\partial H}{\partial \theta }h+\frac{1}{2}\mathrm{tr}\left(H^{-1}\frac{\partial H}{\partial \theta }\right)$$

The following definitions are set:$$g=\underset{\tilde{g}}{\underset{︸}{\frac{\partial r}{\partial a}}}+\underset{J}{\underset{︸}{\frac{\partial s}{\partial a}}}\underset{\hat{g}}{\underset{︸}{\frac{\partial r}{\partial s}}}\;H=\underset{\tilde{H}}{\underset{︸}{\frac{\partial^{2}r}{\partial a^{2}}}}+\underset{J}{\underset{︸}{\frac{\partial s^{T}}{\partial a}}}\underset{\hat{H}}{\underset{︸}{\frac{\partial^{2}r}{\partial s^{2}}}}\underset{J^{T}}{\underset{︸}{\frac{\partial s}{\partial a}}}+\underset{\overset{⌣}{H}}{\underset{︸}{\frac{\partial^{2}s}{\partial a^{2}}}}\underset{\hat{g}}{\underset{︸}{\frac{\partial r}{\partial s}}}$$

The gradient can be rewritten as:(7)$$\begin{array}{c} \frac{\partial ℒ}{\partial \theta }=\sum_{ti}\frac{\partial {\tilde{g}}_{ti}}{\partial \theta }h_{ti}+\sum_{ti}\frac{\partial {\hat{g}}_{ti}}{\partial \theta }{\left[J^{T}h\right]}_{ti}+\frac{1}{2}\sum_{tij}\frac{\partial {\tilde{H}}_{tij}}{\partial \theta }\left({\left[H^{-1}\right]}_{ttij}-h_{ti}h_{tj}\right)+\\ \frac{1}{2}\sum_{tij}\frac{\partial {\hat{H}}_{tij}}{\partial \theta }\left({\left[J^{T}H^{-1}J\right]}_{ttij}-{\left[J^{T}h\right]}_{ti}{\left[J^{T}h\right]}_{tj}\right)+\\ \frac{1}{2}\sum_{ti}\frac{\partial {\hat{g}}_{ti}}{\partial \theta }\sum_{t_{1}t_{2}jk}\left({\left[H^{-1}\right]}_{t_{1}t_{2}jk}-h_{t_{1}j}h_{t_{2}k}\right){\overset{⌣}{H}}_{t_{1}t_{2}jkti} \end{array}$$
where ${\left[H^{-1}\right]}_{ttij}$ denotes the ijth entry in the block tt; J is Jacobian matrix; $h=H^{-1}g$ [38].

### 2.6. Design of Evaluation Function

In Markov modeling, the formulation of the evaluation function affects the generation of strategy. This paper uses Markov model to generate driving behavior and train it according to human driver behavioral characteristics.

At intersections, people often set to execute different driving tasks in different environments. When the environment is relatively safe, people will choose to reach their destination more quickly. When the environment is relatively dangerous, people will first consider safety and will rather sacrifice speed to ensure it. The theory based on utility prospect has two main limitations: (1) The standard model assumes that human beings are risk neutral in their utility function; (2) It assumes that humans do not distinguish scenarios with known outcome probability from scenarios with unknown outcome. The experimental results show that, it is not a good model of human behavior in a risk scenario. It has been established that, human beings not only hate risk, but also have fuzzy aversion. For example, in the decision-making behavior at intersections, regarding the characteristics of human risk aversion modeling, a hierarchical decision model is adopted (Algorithm 2). The upper layer is used as the division of dangerous scenarios, while the lower layer is based on different evaluation functions of safe and dangerous scenarios.

**Algorithm 2** Hierarchical Decision Model
**Input**: The status of the ego vehicle
**If** (Risk>threshold)

      **Then** Execute evaluation function $R_{risk}$

      **Output**: Series dynamics in planning window
**If** (Risk<threshold)

      **Then** Execute evaluation function $R_{safe}$

      **Output**: Series dynamics in planning window

The inverse reinforcement learning of continuous state space is used to learn the human evaluation function from the human driving data. The evaluation function is assumed to be linear and can be expressed as follows:(8)$$R(s,a)=\mu_{1}R_{safety}(s,a)+\mu_{2}R_{time}(s,a)+\mu_{3}R_{comfort}(s,a)$$

Timeliness:(9)$$R_{time}(s,a)=\{\begin{array}{c} {\left(v_{ref}-v_{0}\right)}^{2},if\;v_{0}>v_{ref}\\ v_{ref}-v_{0},if\;v_{0}\leq v_{ref} \end{array}$$
where $v_{ref}$ refers to the desired reference speed and $v_{0}$ refers to the current vehicle speed.

Comfort:(10)$$R_{comfort}(s,a)=jerk+steer$$
where the longitudinal jerk is used to represent the jerk and the steering wheel angle is used to represent the steering angle of the vehicle.

Security:(11)$$R_{safety}(s,a)=-\sum_{i=1}^{n_{s}}{\left(d_{i}\right)}^{2}$$
where *d* represents the distance between two vehicles (m) and $n_{s}$ represents the number of surrounding interactive vehicles.

## 3. Simulation Scenario Verification

### 3.1. Scenario Design

This paper uses the simulation test platform Carla, to build an urban road intersection setup and arrange relevant traffic elements, to simulate the driving behavior of human drivers, in the traffic scenario of urban road intersections. Figure 4 shows the initial interface of Carla.

In this paper, a preset intersection on the Carla simulation platform is selected as the experimental scenario (Figure 5): the other vehicle turns left at the intersection, while the test vehicle goes straight, forming a conflict area of the two vehicles, at the intersection. Since this paper mainly studies the vertical decision-making process of the vehicle, the test vehicle adopts the straight route, while the other vehicle adopts the left-turn route, thus reducing the influence of the horizontal decision-making, component. The trajectory of the other vehicle will be designed according to the acceleration and speed distribution characteristics of the human driver, acting in a real scenario. The longitudinal control of the vehicle is manual and controlled by the accelerator pedal. For this scenario, two types of accidents that occur frequently at intersections, are selected. One is when the other vehicle stops near the conflict area at the intersection and suddenly starts moving. The second accident case is the process, where the other vehicle decelerates sharply, at the intersection.

### 3.2. Experimental Design

The setup is designed as a conflict scenario, where the test vehicle travels straight and the other one turns left. At the beginning, both vehicles stood 30 m away from the intersection. The moment the two vehicles cross the intersection is regarded as the end of time.

Track design of surrounding vehicles: in order to setup the trajectory of surrounding vehicles closer to the driving conditions of the real driver, three driving trajectories, with obvious characteristics from the real vehicle data, were extracted and used as the trajectory input of the opposite vehicle. Based on the speed data of the opposite vehicle, three different speed curves are selected. As shown in the Figure 6, the blue curve represents the case of approaching constant speed and mild passing through the intersection, the black curve represents the scenario of accelerating radically, while passing through the intersection and the red curve represents rapid deceleration and avoidance. In order to extract human sensitive factors to danger, we recruited 7 participants (2 women and 5 men), all of whom had driven at least 2 years of driving experience. The collected driver information is shown in Table 2. We conducted experiments with driving simulator. Driver inputs can be obtained by steering wheel and pedals of the driving simulator, and a total of 74 groups of effective data are collected, of which 56 groups are selected as the test set and 18 groups as the verification set.

According to the verification of the test set (Table 3), it is clear that the coefficient of the collision function, in the dangerous area, will be significantly higher than the coefficient of the collision function, in the safe situation, and in the in-between type of environment. This result shows that, when the risk increases, people will strongly prefer to act in favor of safety, above anything else. In addition, in dangerous scenarios, people’s urge for high speed will be relatively lower. At the intersection, people often slow down to ensure safety, so the experimental results are also in line with this reality.

In the process of learning human behavior, the root mean square (RMS) value of the Risk-sensitive [51] learning system (Figure 7), based on risk preference, is 2.52 km/h, while the error is controlled at about 10% for the driving behavior with an average speed of 17 km/h. The root mean square value of risk-neutral [52,53] learning system is 3.96 km/h, which is significantly higher than that of the risk sensitive learning system, whereas its error exceeds the acceptable range of 15%. The experimental results show that, the risk-sensitive learning system better fits the behavior of human drivers than the risk-neutral learning system, verifying the hypothesis that drivers exhibit the characteristics of risk sensitive behavior, at intersections.

### 3.3. Scenario 1: The Opposite Vehicle Not Giving Way

When there are vehicles passing by in the front, humans usually choose the defensive strategy; that is, stop in front of the traffic flow and wait for all vehicles to pass. In this experiment, the effectiveness of SMDP model is verified by comparing it to conventional strategies. In Figure 8, Figure 9 and Figure 10, the gray-colored curves in the experimental images are the average of the conventional predicted trajectories, and the red line is the average of the human driving trajectories, while the blue line represents the average of the defensive behavior trajectories, whereas the shaded area shows the standard error. The action of this vehicle adopts the discrete action output, with a step size of 0.5 s, providing an output acceleration range of [−2, 0] m/s^2^ at an acceleration interval of 0.5 m/s^2^.

As shown in Figure 8, the predicted vehicle speed curve is roughly consistent with the real scenario, which is in line with the common sense of human behavior. Specifically, in the case of the traffic flow scenario, the choice is to slow down slowly and stop in front of the traffic flow. As shown in Figure 9 and Figure 10, when the opposite vehicle adopts acceleration of 0.25 m/s^2^ or speed of the opposite vehicle is constant, the predicted vehicle speed curve is also roughly consistent with the real scenario, following the common sense of human behavior.

### 3.4. Scenario 2: The Opposite Vehicle Giving Way

In the design of Scenario 2, the other vehicle suddenly decelerates (−0.5 m/s^2^) at a distance from the intersection, but it is impossible to judge whether it can stop before reaching the conflict area, which means that it shows unclear intentions. In this case, both defensive and cooperative strategies are executed. In Figure 11 and Figure 12, the gray line trajectories in the experimental plots are the average of the conventional prediction trajectories, the red line trajectory is the average of the human driving trajectories and the blue line trajectory is the average of the cooperative behavioral trajectories, whereas the shaded area shows the standard error. As shown in Figure 11b, the curve fitting has a large error, while the curves of the cooperative behavior and the conventional behavior coincide. Whether it is the error of the human driver or the comparison to the predicted curve, the error is relatively large. The reason is that, when the acceleration is 0.5 m/s^2^, it is easy for the driver to switch before defensive and cooperative behavior, due to determining the intention of other vehicles, while the behavior strategy proves ambiguous, resulting in a large deviation of the final result. Therefore, the prediction curve error for the driver of the vehicle is large.

As shown in Figure 12, the other vehicle suddenly decelerates sharply with a deceleration of −1 m/s^2^. According to the estimation of the uniform acceleration motion, the other vehicle can stop before reaching the conflict area, which shows that it has a clear intention. In this case, the defensive behavior is executed at the same time. As shown in Figure 12, the comparison of the deceleration curves −1 m/s^2^ and −0.5 m/s^2^ demonstrates that, when the deceleration is −1 m/s^2^, the cooperative behavior fits the human driver data more accurately. This may occur because the intention of the other vehicle is clearer, when the deceleration is −1 m/s^2^, whereas the hierarchical behavior can more accurately fit the human driving behavior. As shown in Figure 12b, the curve fitting error is small, while the gap between the curve of cooperative behavior and the curve of conventional behavior is large. The speed curve predicted by the vehicle is roughly consistent with the real scenario. The vehicle chooses to accelerate, when approaching the conflict area, when facing the scenario of other vehicles evading.

## 4. Real Vehicle Scenario Verification

BYD Tang is proposed to be used as the data acquisition mobile platform, equipped with the sensor layout as shown in Figure 13. This specific platform is also used in [54]. The sensors loaded on the platform are as follows: a Velodyne HDL-32E lidar, an OxTs Inertial+ GNSS/INS suite, Mako cameras and an industrial computer.

The driving action of the vehicle, as generated by the semi-Markov model, is compared to that of the human driver, in a real scenario, in order to verify the effectiveness of the model. This experiment sets two characteristic scenarios for verification: one is the scenario of multiple traffic flow ahead and the other is the scenario of the opposite vehicle turning left to avoid the test vehicle.

### 4.1. Scenario I

When a large number of traffic flows ahead, as the test vehicle drives along, human drivers usually choose defensive strategy, that is, stop in front of the traffic flow and wait for it to pass. During the experiment, the conventional strategies are compared, to verify the effectiveness of the model. The black curve in Figure 14 is the speed curve of the actual vehicle, while the red curve is the predicted speed curve. The action of the test vehicle adopts discrete action output, with a step of 0.5 s and the range of output acceleration is [−2,0] m/s^2^, at an acceleration interval of 0.5 m/s^2^.

As shown in Figure 14, the predicted speed curve of the vehicle is roughly consistent with the real scenario. When facing the scenario of traffic flow, the vehicle chooses to slow down and stop in front of the traffic flow. Compared to conventional behavior, defensive driving behavior shows three characteristics:(1)Defensive behavior, similarly to human behavior, has a turning point in the middle of deceleration. The turning point of the vehicle speed curve decline is roughly consistent, indicating that human beings follow a rapid deceleration process, when approaching the conflict area. This shows that the learned defensive behavior is based on the defensive characteristics of human beings;(2)Defensive behavior actually achieves the defense effect by setting a safe area. Compared to conventional behavior, it still does not slow down to 0, when approaching the conflict area, which still poses a great risk of collision within the conflict area, indicating that setting a safe area can indeed achieve the effect of ensuring safety.

As shown in Figure 15, the blue vehicle is the test vehicle (ego vehicle) and the green vehicle is the opposite vehicle. Compared with defensive behavior, conventional behavior does not learn such a behavior through the reward function. Finally, it is quite close to other vehicles and is in an extremely dangerous situation.

### 4.2. Scenario II

When there is a vehicle waiting to pass on the left, human drivers usually choose cooperative strategy, to accelerate driving.

As shown in Figure 16, the predicted speed curve of the vehicle is roughly consistent with the real scenario. When facing the scenario of avoiding the opposite vehicle, the test vehicle chooses to accelerate, when approaching the conflict area. Compared to conventional driving behavior, cooperative driving behavior exhibits three characteristics:(1)Similar to human behavior, cooperative behavior has a turning point in the middle of acceleration. The turning point of the rising speed curve of the vehicle is roughly consistent, indicating that human beings follow a rapid acceleration process, when approaching the conflict area. It reflects the human response to the other vehicle avoidance behavior. This shows that, the learned cooperative behavior is based on the cooperative characteristics of human beings;(2)In fact, cooperative behavior means to leave the conflict area after 3 s. Conventional behavior is not based on the reward function. On the contrary, in order to avoid being too close to the obstacle, the vehicle is in the state of deceleration and gives way to the opposite vehicle, so it does not try to understand the intention of the opposite vehicle;(3)As shown in Figure 17, the blue vehicle is the test vehicle (ego vehicle) and the green vehicle is the opposite vehicle. The conventional behavior, compared to cooperative behavior, is not developed using the reward function, after constraining the vehicle to leave the conflict area for 3 s. Finally, it is quite far away from the opposite vehicle. In the environment of large traffic flow, parking arbitrarily at the intersection, in a way that other drivers do not understand, will put the vehicle in an extremely dangerous situation.

To verify the effectiveness of our method, one of the state-of-the-art IRL methods presented in [40] is selected for comparison. To measure the similarity between human-like systems and human drivers, Human Likeness (HL) is defined by [40]. HL is the L2 norm between the position at the end of the ground truth trajectory and that of the closest prediction among the three most likely trajectories.
(12)$$HL=\min{\left\{{\left\|{\hat{\xi }}_{i}(L)-\xi_{gt}(L)\right\|}_{2}\right\}}_{i=1}^{3}$$
where ${\hat{\xi }}_{i}(i=1,2,3)$ are the selected trajectories with the highest probabilities, $\xi_{gt}$ is the ground-truth trajectory by the human driver, and L is the end of the time horizon. Applying this concept to this paper, we can get human likeness as follows:(13)$$HL={\left\|\hat{\xi }(L)-\xi_{gt}(L)\right\|}_{2}$$
where $\hat{\zeta }$ is the trajectory that generated by the proposed method.

Another indicator, the root mean square error (RMSE), can be used to measure the stability of different methods. A higher RMSE means that the method is less stable. The RMSE here can be calculated by:(14)$$RMSE=\sqrt{\frac{\sum d_{i}^{2}}{n}}$$
where $d_{i}$ is the deviation of a set of measurements from the true value, and i=1,2,3,….

As can be seen in Figure 18a, the HL of our method are very low, mostly around 2, except for the fourth set of data. Since the fourth set of data is a scenario in which the opposite vehicle decelerates to stop and gives way, and the ego vehicle accelerates to pass through the intersection, there is no other restriction on the speed of ego vehicle if the speed limit is not exceeded, so the error is relatively large. As shown in Figure 18b, the RMSE of the proposed method (2.356) can be reduced by 12.1% compared with the RMSE of general modeling (2.681) in [40], which shows that our proposed method is closer to human behavior and has better stability. In addition, the scenario in [40] is on a straight road, and only interactions with vehicles in the same direction are considered. While this paper focuses on the intersection with vehicles moving in different directions. Decision making at the intersection is much more complicated than that in [40].

## 5. Conclusions

This paper is based on the “BYD Tang” autonomous driving platform of the Intelligent Vehicle Research Institute of Beijing Institute of Technology to obtain human driving data. This paper analyzes the driving characteristics of human drivers through actual driving data and solves the key problems faced by autonomous vehicles. The contributions of this paper are listed below:(1)A human-like behavior decision-making method is proposed for decision-making in intersection scenarios. It is developed based on inverse reinforcement learning and semi-Markov model. The proposed method can learn well from demonstration trajectories of human experts, generating human-like trajectories;(2)The proposed model combines driving intention of other vehicles and ego vehicle, and uses the semi-Markov model to directly correspond the intention to the driving strategy, which adapts to the intention ambiguity of other vehicles in the real scene. Meanwhile, it abstracts the interaction behavior between other vehicles and ego vehicle into defensive behavior and cooperative behavior, and takes different behaviors for different scenarios;(3)Compared to traditional methods, the proposed method can learn well from human demonstration data, and is more in line with human driving behavior. The analysis results show that the method also has good stability, in addition with accuracy.

Although this work has achieved some positive results, there are still some areas that can be further improved. First, the discretization of action space could lead to the error of behavior results. Second, this paper focuses the test vehicle and its surrounding vehicles, without considering other traffic participants, such as pedestrians, non-motor vehicles and so on. In future studies, other traffic participants will be considered to establish a more complete scenario model. At the same time, sampling methods will be considered, so as to reduce the error caused by discretization.

## Figures and Tables

**Figure 1 sensors-22-04500-f001:**
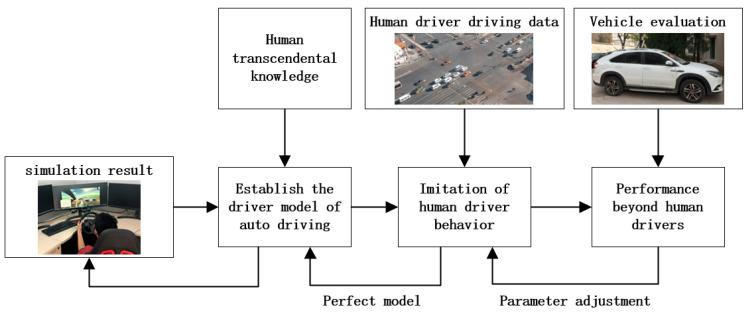
Intersection decision framework.

**Figure 2 sensors-22-04500-f002:**
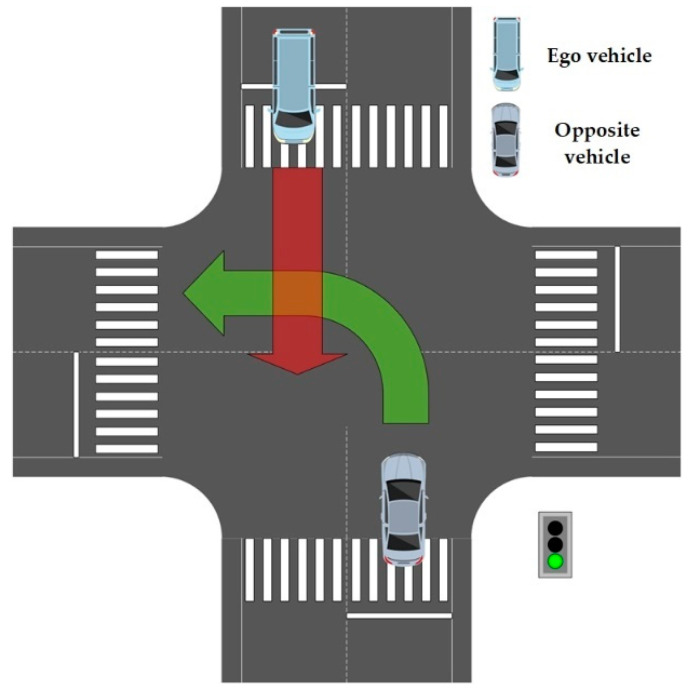
Potential collision area.

**Figure 3 sensors-22-04500-f003:**
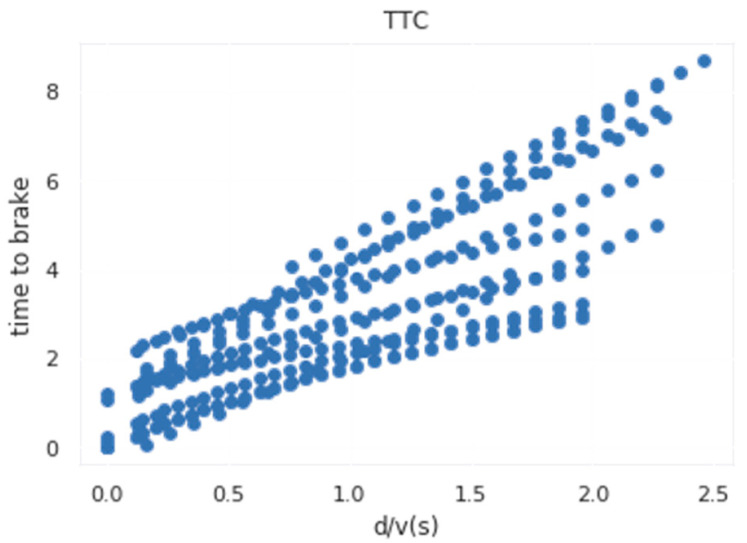
Relationship between braking time and *d/v*.

**Figure 4 sensors-22-04500-f004:**
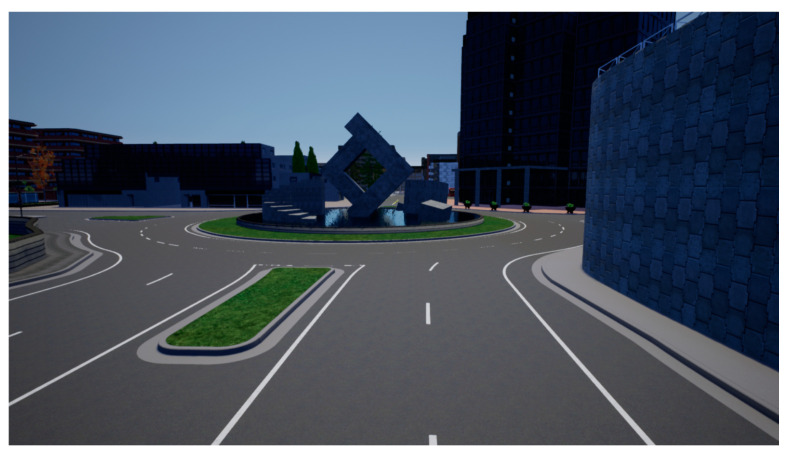
Carla initial interface.

**Figure 5 sensors-22-04500-f005:**
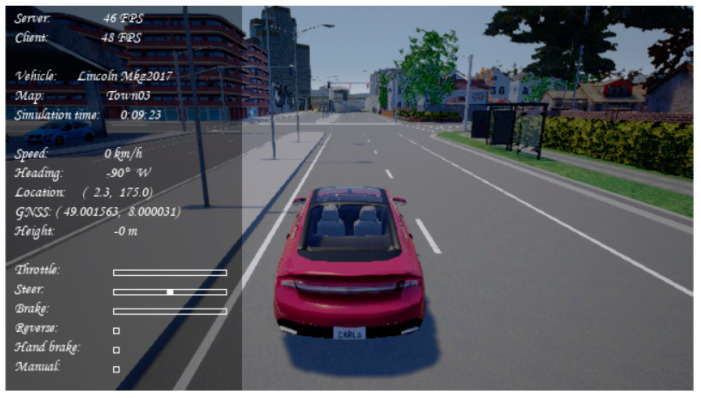
Carla intersection scenario.

**Figure 6 sensors-22-04500-f006:**
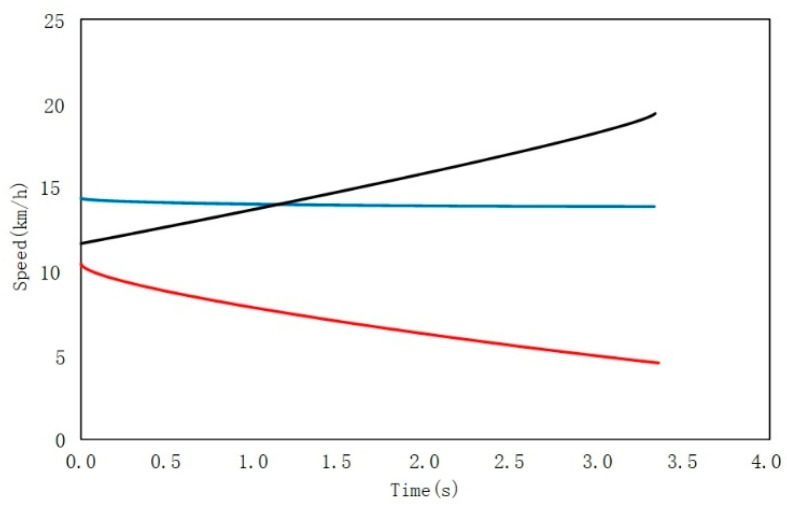
Semi-Markov model.

**Figure 7 sensors-22-04500-f007:**
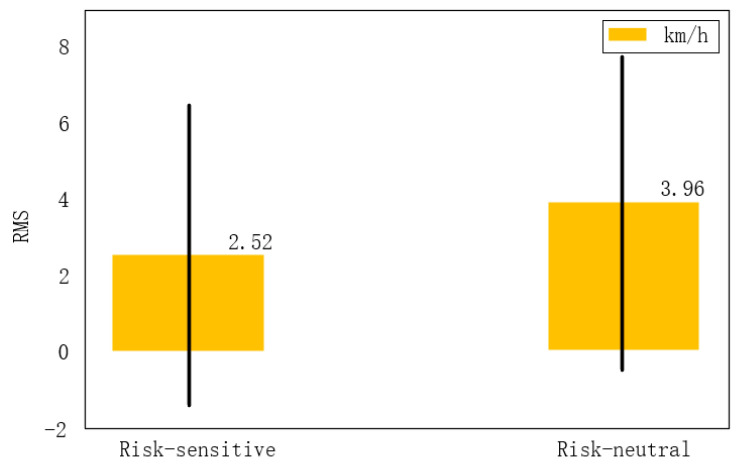
The opposite vehicle speed trajectory design.

**Figure 8 sensors-22-04500-f008:**
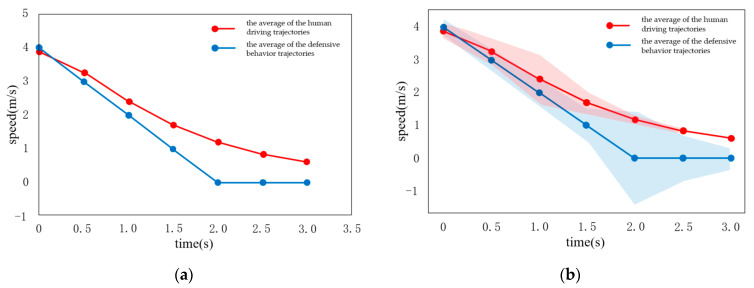
When the deceleration of other vehicles is 0.5 m/s^2^. (**a**) Conventional and cooperative speed curves; (**b**) Comparison to real vehicle data.

**Figure 9 sensors-22-04500-f009:**
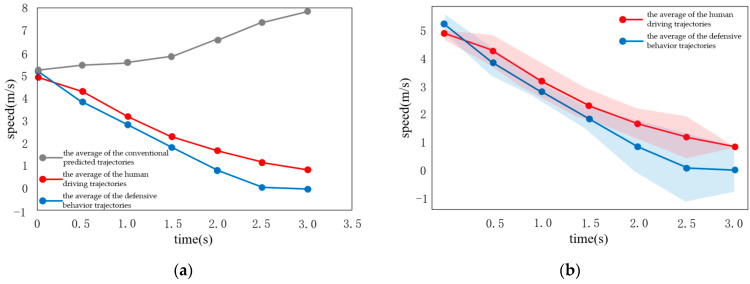
Situation where the opposite vehicle is not giving way. (**a**) Conventional and cooperative velocity curves; (**b**) Comparison to real vehicle data.

**Figure 10 sensors-22-04500-f010:**
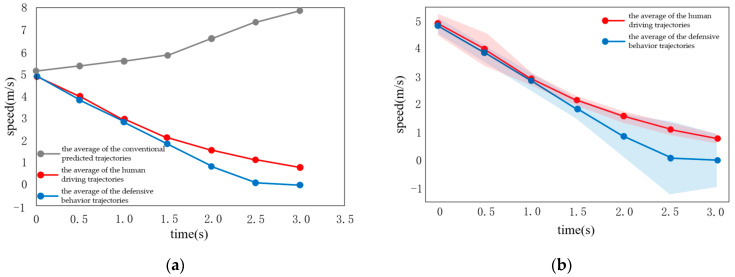
Other vehicles at constant speed. (**a**) Conventional and cooperative velocity curves; (**b**) Comparison with real vehicle data.

**Figure 11 sensors-22-04500-f011:**
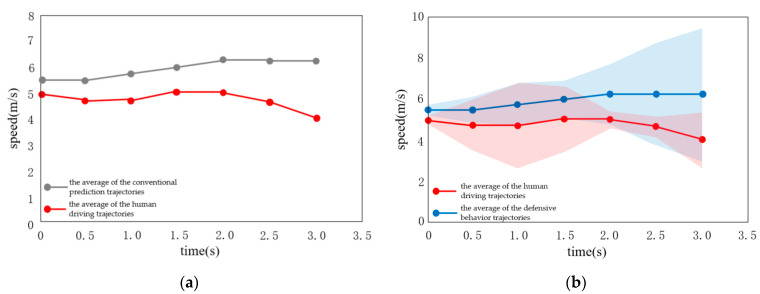
Deceleration of other vehicles is −0.5 m/s^2^. (**a**) Conventional and cooperative speed curves; (**b**) Comparison to real vehicle data.

**Figure 12 sensors-22-04500-f012:**
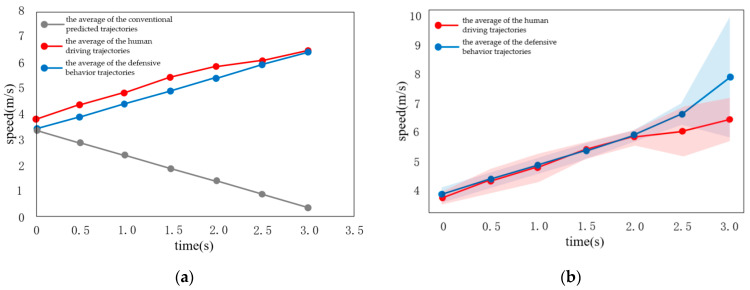
Situation where the opposite vehicle is avoiding. (**a**) Conventional and cooperative velocity curves; (**b**) Comparison to real vehicle data.

**Figure 13 sensors-22-04500-f013:**
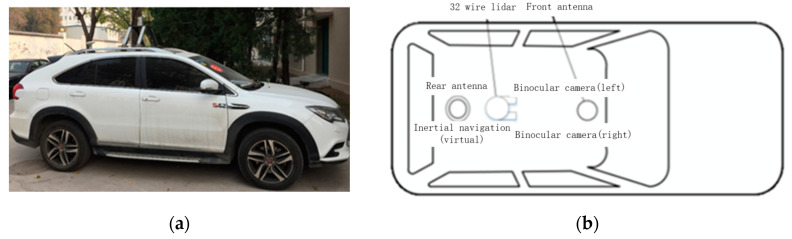
Sensor arrangement. (**a**) BYD Tang test vehicle; (**b**) Sensor layout plan.

**Figure 14 sensors-22-04500-f014:**
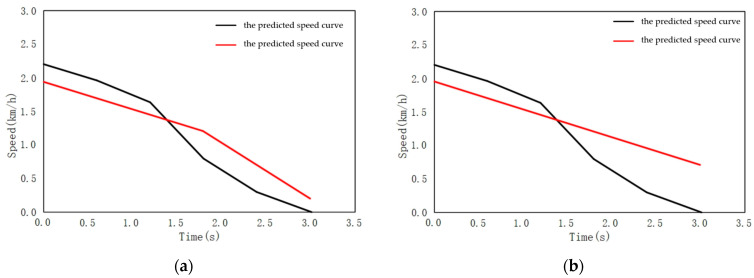
Comparison of prediction curves of defensive behavior and conventional behavior. (**a**) Predicted speed curve for defensive behavior; (**b**) Predicted speed curve for conventional behavior.

**Figure 15 sensors-22-04500-f015:**
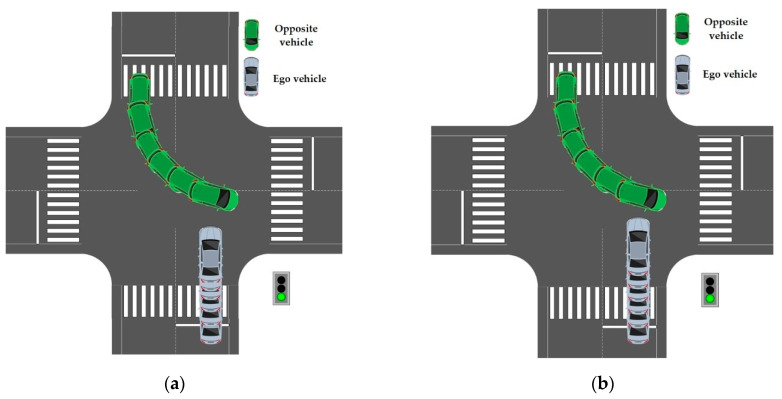
Comparison of produced trajectory between defensive algorithm and conventional algorithm. (**a**) Trajectory of defensive behavior; (**b**) Trajectory of conventional behavior.

**Figure 16 sensors-22-04500-f016:**
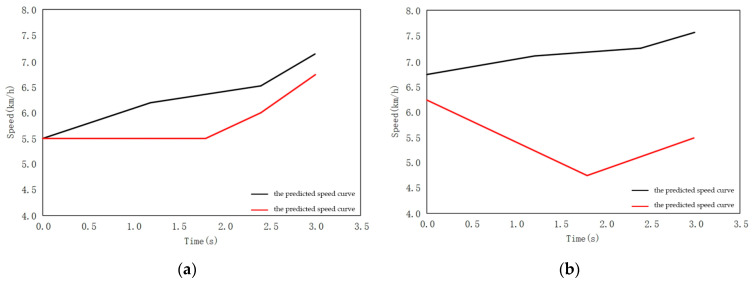
Comparison of prediction curve between interactive algorithm and conventional algorithm. (**a**) Predicted speed curve for interactive behavior; (**b**) Predicted speed curve for conventional behavior.

**Figure 17 sensors-22-04500-f017:**
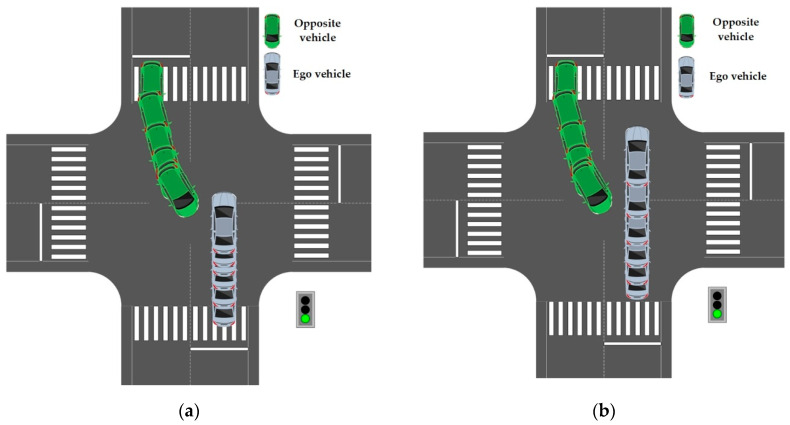
Comparison of motion trajectories between interactive algorithm and conventional algorithm. (**a**) Motion trajectory of interactive behavior; (**b**) Motion trajectory of conventional behavior.

**Figure 18 sensors-22-04500-f018:**
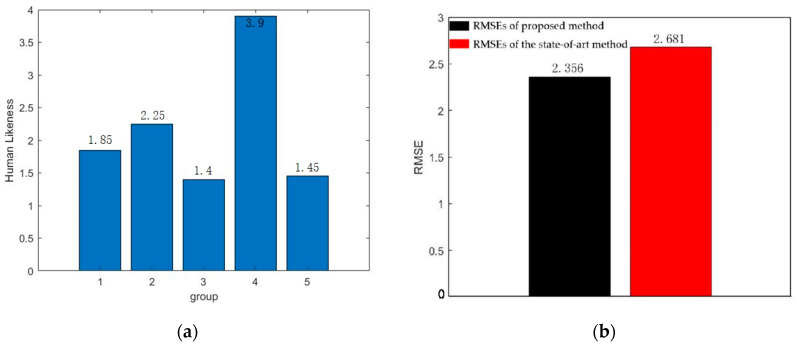
Comparison with the state-of- art method. (**a**) Human likeness of the proposed method; (**b**) RMSEs of our proposed method and the state-of-art method.

**Table 1 sensors-22-04500-t001:** Correlation analysis.

Category	$T_{break}$	v	d	d/v
$T_{break}$	1.000	0.834	−0.104	0.9178
v (m/s)	0.834	1.000	0.038	0.978
d (m)	−0.104	0.038	1.000	−0.054
d/v	0.918	0.978	−0.054	1.000

**Table 2 sensors-22-04500-t002:** Driver’s information participating in the experiment.

Average of Age (Years)	Standard Deviation of Age	Average of Driving Experience	Standard Deviation of Driving Experience (Years)
24.43	1.40	4.43	1.76

**Table 3 sensors-22-04500-t003:** Evaluation functions under three states.

Situation	$R_{safety}$	$R_{time}$	$R_{comfort}$
Danger	47.677	0.198	1
Security	4.56	13.09	1
Risk neutral	9.45	2.09	1

## Data Availability

Not applicable.

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
