# Peer review of "Human-like Decision Making for Autonomous Vehicles at the Intersection Using Inverse Reinforcement Learning"

_sensors, 2022, doi:10.3390/s22124500_

Round 1

Reviewer 1 Report

Dear authors

The revision has been improved greatly. The paper becomes readable and understandable. The authors replied the comments carefully and politely.

I still have some comments about the revison:

  1. line25: "Google is far ahead in the field of automatic driving". Due to the scientific research paper, subjective appraisal to specific complany should be avoided.
  2. As mention in previous review, the items used in the paper should be exact. For example, what is "driving logic (line 44)" that is same content as "driving behavior (line 55)"?
  3. Figure 1: 2 section's title is "Framework of decision model", 2.1 section's title is "Intersection decision framework", and figure 1's label is "intersection decision process", they are same concept? If yes, please unify them; if no, please show their difference.
  4. Authors give the legends in the paper. If these legends are added in figures, these figures would become more understandable.
  5. Authors used many paragraphs to explain the Maximum entropy inverse reinforcement learning, Maybe it is needed, but are symbols in these equations really exact? For example, jerk and steer in (10) line 27.
  6. Authors added the table 2 to show the participants' profile based on my comment. Average and sd of Age and Experience are enough instead of individual age...
  7. Figure 9. Authors show that risk-neutral learning system is SIGNIFICANTLY HIGHER THAN risk sensitive one. Please explain what kind of statistical analysis has been used, and which result support the conclusion?
  8. Figure 17. (a) The trajectory of ... and (b) Trajectory of ... please unify the lables.

Reviewer 2 Report

Planning and decision play an important role in the field of self-driving vehicles. The authors proposed a human-like decision making method at the intersection based on inverse reinforcement learning. Comprehensive experiments are conducted on both the CARLA simulator and a real vehicle. However, there are some problems in this paper.

(1) Contributions of this paper are obscure and weak.

(2) To few state-of-the-art methods in the field of end-to-end autonomous driving are introduced in this paper.

(3) The contents, Lines 192-195, lack theoretical basis. There is no literature or experimental data to support it.

(4) There are some errors in some statements of the paper, such as the sentence in Line 295. The longitudinal control is the accelerator rather than steering wheel. The authors need to carefully check the whole paper.

(5) The definition of some terms in this paper is not clear. For example, Line 332, the vocabulary of risk-sensitivel learning system and risk-neutral learning system in the following section are presented for the first time. However, they have not been mentioned before, and there is no related reference, which leads to a misunderstanding of the related experiments.

(6) The data in the figures in the experiment part are not clear. Moreover, some chart information is not explained in detail in the article, such as Tab. 1 and Fig. 3.

(7) The role of reverse reinforcement learning is not explained in Section 2.5.

Reviewer 3 Report

Scope and aim of the manuscript are contemporary and very significant especially in the model development of  the human decision making and in analyze of autonomous vehicle movement.

In order to develop appropriate model of human like decision making for autonomous vehicles at the intersection used Inverse Reinforcement learning method as well as Semi-Markov Decision Process. Three different model of human behavior were analyzed: conventional, cooperative and defensive.

Manuscript is clear and well-structured.

Used references in introduction and literature review are mostly recent publication.

Also, misunderstandings exist in manuscripts:

Page 9:

- In explanation of equation (9), difference between vref and vo should be more clear than it is given.

- Which physical value represent jerk and steer? Steering Wheel angle, torque, camera signal, LIDAR signal, etc?

Page 10:

- Describe effective data used as input data as well as conducted experiment.

Page 12:

Scenario description is not clear enough. For example, in line 355 and 356: “As shown in Figs. 11 and 12, when the acceleration is 0.25m/s2 and the speed is constant, the predicted vehicle speed curve is also roughly consistent with the real scenario, following the common sense of human behavior”.

It should be point out that the first vehicle accelerate and speed of the second vehicle is constant.

Obtained results should be compared with recently published results. Benefits of applied methods and obtained results should be emphasized.

Round 2

Reviewer 2 Report

The authors have carefully modified this paper according to the reviewers’ comments. However, following comments can help further improve this paper.

1. It is suggested to highlight the contributions with items.

2. The authors could use algorithms to replace the flowcharts such as Figs 4 and 5 to make this paper more legible.

3. Which is the most similar method to the proposed method? The differences between the proposed method and existing methods are not clearly described.

4. Too little description about the motivation is shown, which weakens the scientificity of this paper.

5. This paper lacks convictive comparative experiments with state-of-the-art methods of end-to-end autonomous driving such as Ref. 30-34 when verifying the validity of the proposed method.

6. The texts and lines in Figs. 8, 16, and 8, etc. are deckle-edged. The authors need to use more clear figures.

Reviewer 3 Report

Dear Author,

I have reviewed revised manuscript. It is improved significantly and I recommend Editor to publish it.

Author Response

Dear reviewer3:

We would like to thank you for your recognition and recommendation of our paper.

This manuscript is a resubmission of an earlier submission. The following is a list of the peer review reports and author responses from that submission.

Round 1

Reviewer 1 Report

Decision is one of the most important parts for autonomous driving vehicles. A human-like decision making method for autonomous vehicles at the intersection based on inverse reinforcement learning is proposed in this paper and achieves apparent good results. However, there are some problems in this paper.

  1. The inverse reinforcement learning method is included in the title, but it is not the main utilized method in this paper.
  2. The three contributions claimed by the authors are not clear.
  3. In term of writing, this paper is more like an engineering report rather than a research paper. It costs too many words to introduce the existing method, which further weakens the innovation of this paper.
  4. This paper could be shortened.
  5. The introduction is too simple and lacks state-of-the-art methods. Furthermore, it lacks logicality in writing.
  6. Figures in this paper are not sightly, which cannot express the authors’ core idea.

Reviewer 2 Report

This paper is about driving evaluation about how close unmanned vehicle behavior resembles human drivers. This paper is well written and can be accepted as it is.

Reviewer 3 Report

This paper focused an important issue for "driverless car" in more complex urban traffic environment. Authors tried to describe their study thoroughly, and explained the study in a complete manner (background, motivation, method, results and conclusion).

However there are many crucial problems in this paper.

  1. I can not totally agree with the opinion in "Introduction" - too conservative driving behavior seems "not recommended" in intersection -.  Though conservative behavior is not "efficient", but not exclusive to "driving safety". 
  2. "driverless vehicles" and "autonomous vehicles" are used in the paper, but there is not clear definitions to define driver roles. In another words, "driverless vehicles" is not completely equal to "autonomous vehicles".
  3. When I read the paper, I cannot judge what are previous studies' conclusions or suggestions, and what are original opinions in this study.
  4. there is too long description of existing models or methods.
  5. authors mentioned several times about the importance of traffic participants' uncertainty in intersection based on previous studies. However, they did not mention their efforts on this point in their study at all. Please give a reasonable explanation why this study did not consider it.
  6. Framework of cross decision model. It is really confused section. There is not logical explanation. For example, why have items of "behavioral decision", "scene evaluation", "TTC", "Semi-Markov Model", "design of evaluation function" been explained? Did all of the contents in this section propose originally in this study? if Yes, please mention it clearly, if no, please mention what is(are) citations and what is(are) proposed.
  7. Figures in this paper are confused to understand what authors wanted to express. For example, Figure 1 concluded 3 main parts, and several sub-parts in each main parts, but no any explanation was given to show their relations; there are neither inputs nor output in this figure. in Figure 3, 21 and 24, there is no note to show which one is ego.
  8. the paper is lack of  ethical approval (Important!)
  9. there were only 7 individuals to participate the data collection. no ages and driving experience have been mentioned, so I am doubtful its validness.
  10. There are several figures which it seems not necessary, like figures 19 22.
  11. Please mention the limitations of the study.

minor points:

  1. Abbreviations should be mentioned only at first  appearance,  Line 191, Markove process model (POMDP) and Line 137 POMDP.
  2. ...(line 124) Zhang et al. showed in HIS paper,... Not HIS, but THEY.
  3. ...(line 182) Song et al.... (line 184) HE used GMM and HMM... not HE but THEY.
  4. line 182: learn human reasoning ... not LEARN but LEARNING.
  5. Please mention what the paper indicated or whether the proposed method is effecitive.